# Enhancement of Agricultural Policy/Environment eXtender (APEX) Model to Assess Effectiveness of Wetland Water Quality Functions

**Amirreza Sharifi** [1],*, **Sangchul Lee** [2,3],*, **Gregory W. McCarty** [3] , **Megan W. Lang** [4],
**Jaehak Jeong** [5] , **Ali M. Sadeghi** [3] **and Martin C. Rabenhorst** [2]

[1] Department of Energy and Environment, Government of the District of Columbia,
   Washington, DC 20002, USA
[2] Department of Environmental Science and Technology, University of Maryland,
   College Park, MD 20740, USA; mrabenho@umd.edu
[3] Hydrology and Remote Sensing Laboratory, USDA-ARS, Beltsville, MD 20705, USA;
   Greg.McCarty@ars.usda.gov (G.W.M.); asadeghi523@gmail.com (A.M.S.)
[4] Branch of Geospatial Mapping and Technical Support, National Wetlands Inventory,
   Falls Church, VA 22041, USA; megan_lang@fws.gov
[5] Blackland Research and Extension Center Texas A&M, Agrilife Research, Temple, TX 76502, USA;
   jjeong@brc.tamu.edu
*  Correspondence: amirrezasharifi@gmail.com (A.S.); sangchul.lee84@gmail.com (S.L.)

**Abstract:** The Agricultural Policy/Environmental eXtender (APEX) model has been widely used to assess changes in agrochemical loadings in response to conservation and management led by US Department of Agriculture (USDA). However, the existing APEX model is limited in quantification of wetland water quality functions. This study improved the current model capacity to represent wetland water quality functions by addition of a new biogeochemical module into the APEX model. The performance of an enhanced APEX model was tested against five observed outgoing water quality variables (e.g., sediment, organic N, $NO_3$, $NH_4$ and $PO_4$) from a wetland within the Eastern Shore of Maryland. Generalized Likelihood Uncertainty Estimation (GLUE) was implemented to assess model uncertainty. The enhanced APEX model demonstrated that it could effectively represent N and P cycling within the study wetland. Although improvement of model performance was limited, the additions of wetland biogeochemical routines to the APEX model improved our understanding of inner mass exchanges within N and P cycling for the study wetland. Overall, the updated APEX model can provide policymakers and managers with improved means for assessment of benefits delivered by wetland conservation.

**Keywords:** APEX; USDA-CEAP; wetland water quality benefit; biogeochemical module; WetQual

## 1. Introduction

The US Department of Agriculture (USDA) Conservation Effects Assessment Project (CEAP) is a multi-agency effort with an overarching objective of assessing and quantifying the effects and effectiveness of USDA's conservation programs and practices in agricultural landscapes across the United States [1]. Among the many conservation programs led by USDA, the Natural Resources Conservation Service (NRCS) Wetlands Reserve Enhancement Program (WREP) (formerly Wetlands Reserve Program—WRP) specifically targets wetlands and offers landowners monetary incentives and technical support to restore, conserve and enhance wetlands and improve wildlife habitat on their property [2]. WREP sponsors a wide range of conservation practices in natural (e.g., riparian areas that

linked protected wetlands), restored (previously restored wetlands that need long-term protection) and historic wetlands (e.g., farmed wetlands or prior converted croplands [2]).

CEAP-Wetlands is one of five thematic national CEAP components, specifically focusing on quantifying wetland ecosystem services (e.g., water quality, flood control or biodiversity) and interpreting effects/effectiveness of conservation practices on wetland functions and services through field research, data collection, model development and model application [3]. To better quantify effects of wetland conservation, there was a pressing need for a model that could be used to assess effectiveness of wetland restoration for water quality improvements, including contaminant/sediment amelioration, nutrient management and surface runoff/floodwater management.

The desired model would be able to capture wetland processes and would cover both upland and wetland biogeochemistry, so that newly restored wetlands and associated upland could be covered in one model. The resulting wetland model could also be used for scenario analysis and for targeting wetland restoration sites that would provide maximum benefits in terms of water quality improvements.

The Riparian Ecosystem Management Model (REMM) and Soil and Water Assessment Tool (SWAT) have commonly been employed to examine wetland water quality functions; both models however are limited by their oversimplified representation of wetland biogeochemical processes [4,5]. In addition, REMM was designed to model riparian wetlands and is therefore not suitable for assessing restored wetlands, which are frequently distant from the stream network. The SWAT model was developed to assess watershed scale hydrology, not field scale hydrology where wetland restoration and management are conducted.

As a part of the CEAP-Cropland Assessment, scientists use the Agricultural Policy/Environmental eXtender (APEX) model to estimate field-level impacts of agricultural conservation practices on crop yield, soil, nitrogen (N), phosphorus (P), carbon (C) and pesticide dynamics [6–8]. The capacity to simulate conservation practices that include establishment, enhancement or restoration of wetlands has not been well developed in APEX, partly due to the inherent complexity and dynamic nature of wetland water and nutrient cycles, the past deficit of sufficient field data to quantify these cycles and the difficulty in determining spatial and temporal presence of wetland hydrology in an agricultural landscape [9,10].

The objective of this study was to introduce an enhanced APEX module that has the capacity to simulate biogeochemical cycling in ponded wetlands. We adopted an existing wetland biochemical processes from the WetQual model [11] and added it to the base APEX model. The performance of an enhanced model was tested using field collected data from a small restored wetland on the Eastern Shore of Maryland where N, P and sediment loads were monitored for approximately 2 years. We first describe the background and methodology for development of a biogeochemical module for N and P cycling within APEX and then evaluated model performances.

## 2. Materials and Methods

### 2.1. Biochemistry in APEX Model

APEX and its earlier counterpart Environmental Policy Integrated Climate (EPIC), are widely tested, comprehensive agro-ecosystem models capable of simulating the growth of crops grown in complex rotations and management operations, such as tillage, irrigation, fertilization and liming [12,13]. APEX and EPIC share various modules; in particular, they share the upland biogeochemistry modules that are responsible for simulating N, P and C cycles in upland soils. These modules follow a modified approach used in the Century model for simulation of soil organic matter [14].

Century's approach, splits soil organic C and N into three compartments that have different turnover times ranging from days or weeks for biomass to hundreds of years for passive organic matter. These compartments in APEX vary in size and function and include: microbial biomass, slow humus and passive humus (Figure 1).

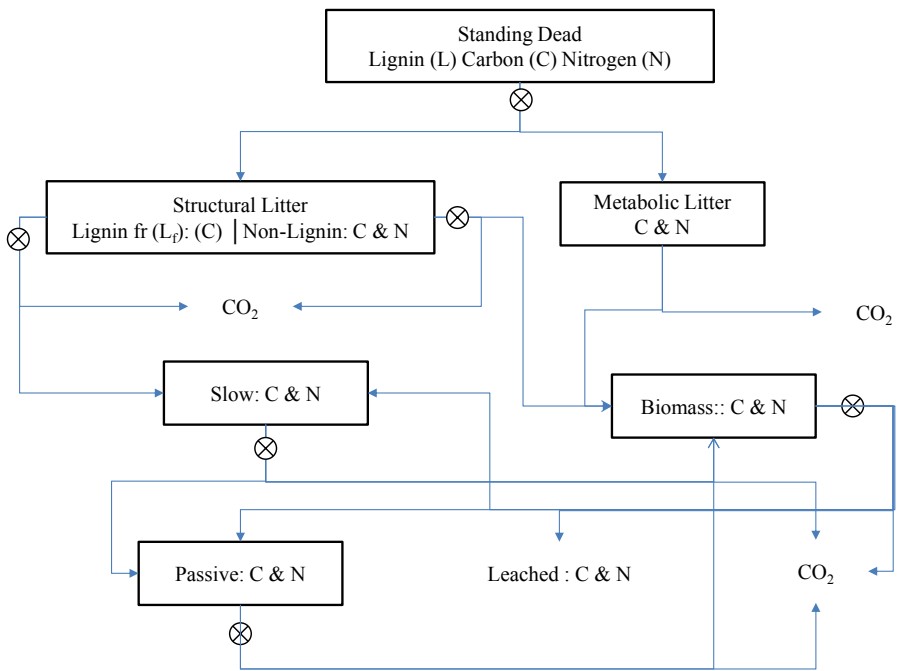

**Figure 1.** Carbon (C) and nitrogen (N) pools in APEX/EPIC (redrawn from Izaurralde, et al. [12]).

Izaurralde, et al. [12] reported four main distinctions between the Century and EPIC/APEX models regarding organic transformations. These distinctions concern relationships that explain leaching of organic material, transformation rates of organic pools and lignin and biogeochemistry of litter on top of the soil layer.

In APEX, the soil organic C and N pools are split into three compartments, including microbial biomass, slow humus and passive humus [13]. The soil organic compartments differ in size, function and turnover times, which range from days or weeks for biomass to hundreds of years for passive organic matter. Carbon and N can also be leached or lost in gaseous forms. In addition to soil organic pools, there are other litter related pools in each soil layer, including C and N in structural litter, metabolic litter and standing dead biomass. Soil inorganic N pools include total ammonium ($NH_4$) and nitrate ($NO_3$).

The transformations between various belowground organic pools are mediated by various factors, including substrate-specific rate constants (rate of potential transformation under optimal conditions), temperature, soil oxygen content, soil water content, lignin content and soil texture. As transformations between organic and inorganic pools can happen in both directions (i.e., mineralization or immobilization) on any given days, the APEX model follows a multi-step process to estimate actual daily transformations from/to each organic pool. The sequence of steps taken to calculate mineralization, immobilization and other transformations between various soil organic and litter pools on any given day are described in the following sections.

### 2.1.1. Potential C and N Transformation

Potential daily organic and litter pool transformations are calculated based on substrate-specific rate constants, temperature, water content, lignin content and soil texture. For instance, potential transformation of C in structural litter (LSCTP) on the surface and subsurface is calculated as a function of the C content in structural litter (LSC), the rate of potential transformation of structural litter under optimal conditions (LSR), a control of the lignin fraction of structural litter (XLSLF) and a combined factor (CS) expressing the effects of temperature, soil water content, oxygen and tillage on biological processes:

$$LSCTP = LSC \ \times LSR \times LSC \times XLSLF \times CS \tag{1}$$

Potential transformation of N in structural litter (LSNTP) is calculated as:

$$LSNTP = LSCTP \times (\text{mass of N in structural litter} \div \text{mass of C in structural litter}) \qquad (2)$$

### 2.1.2. Potential Transformations are Allocated to Receiving Pools

Potential transformations calculated in the last step are allocated to receiving pools. For instance, during its transformation, metabolic litter is partitioned into $CO_2$ (55%) or biomass (45%).

### 2.1.3. N Demand for Each Potential Transformation is Estimated

The demand for N is established by potential C transformation of the source pool and the N/C ratio of the receiving compartment. For instance, N demand (PN) of Metabolic Litter $\rightarrow$ Biomass transformation is calculated as:

$$PN = LMCTP \times 0.45 \times NCBM \qquad (3)$$

where LMCTP is potential transformation of C in metabolic litter (kg ha$^{-1}$ day$^{-1}$) and NCBM is N/C ratio of biomass.

### 2.1.4. Actual C and N Transformations

Actual C and N transformations are calculated based on N supply (available from each potential transformation) and N demand (established by potential C transformation of the source compartment and N/C ratio of the receiving compartment). If the N available exceeds the combined demand in all receiving compartments, the potential transformation then becomes the actual transformation. Thus, the calculated N and C flows are added to the receiving compartment and subtracted from the source compartment. The excessive available N (N supply – N demand) is then added to the $NH_4$ pool (mineralization). If amount of N available falls short relative to combined demand in all receiving compartments, then actual transformations are smaller than estimated potential transformations. Actual transformations are calculated as follows:

$$\text{Actual transformation} = \frac{\text{Potential transformation} \times \text{ total available N}}{\text{total N demand}} \qquad (4)$$

In this case, all available N is consumed during transformation (immobilization).

In the APEX model, nitrification is estimated as a function of available $NH_4$, pH, soil moisture and temperature. Volatilization, loss of ammonia to the atmosphere, is estimated simultaneously with nitrification as a function of temperature and wind speed. Denitrification is estimated as a function of available nitrate, temperature and soil carbon content and only occurs when soil is saturated above a certain content (soil water factor >0.95, see Williams, et al. [13]).

### 2.2. Description of the Newly Added Wetland Module

After careful consideration and consultation with APEX developers, it was decided that modification of original APEX biogeochemistry modules for inclusion of wetland biogeochemistry was not practical. Instead, addition of new code to include extra soil and water layers and proper methodology to address flooded wetland biogeochemistry was deemed appropriate. For this purpose, a new module was added to APEX, which implements a methodology for wetland nutrient cycling and is attached to the reservoir component of APEX. The methodology of the new module was mainly adopted from the WetQual model [11,15], a well verified and detailed process-based model for flooded wetland biogeochemistry.

Figures 2 and 3 depict various transport mechanisms and loss pathways for nitrogenous species and phosphorus in both free-water and sediment compartments of the newly added module in APEX. As mentioned earlier, many of the concepts described in this section were adopted from the WetQual

model [11]. The main difference between the APEX wetland module and the WetQual model is the existence of a thin aerobic soil layer in WetQual on top of the anaerobic soil column. This layer was eliminated in the APEX wetland module to save computational capacity due to its minimal impact on overall model results. Mass of the constituents in each nutrient pool (N and P pools presented in Figures 2 and 3) is accounted for in mass balance ordinary differential equations that are presented in the following sections.

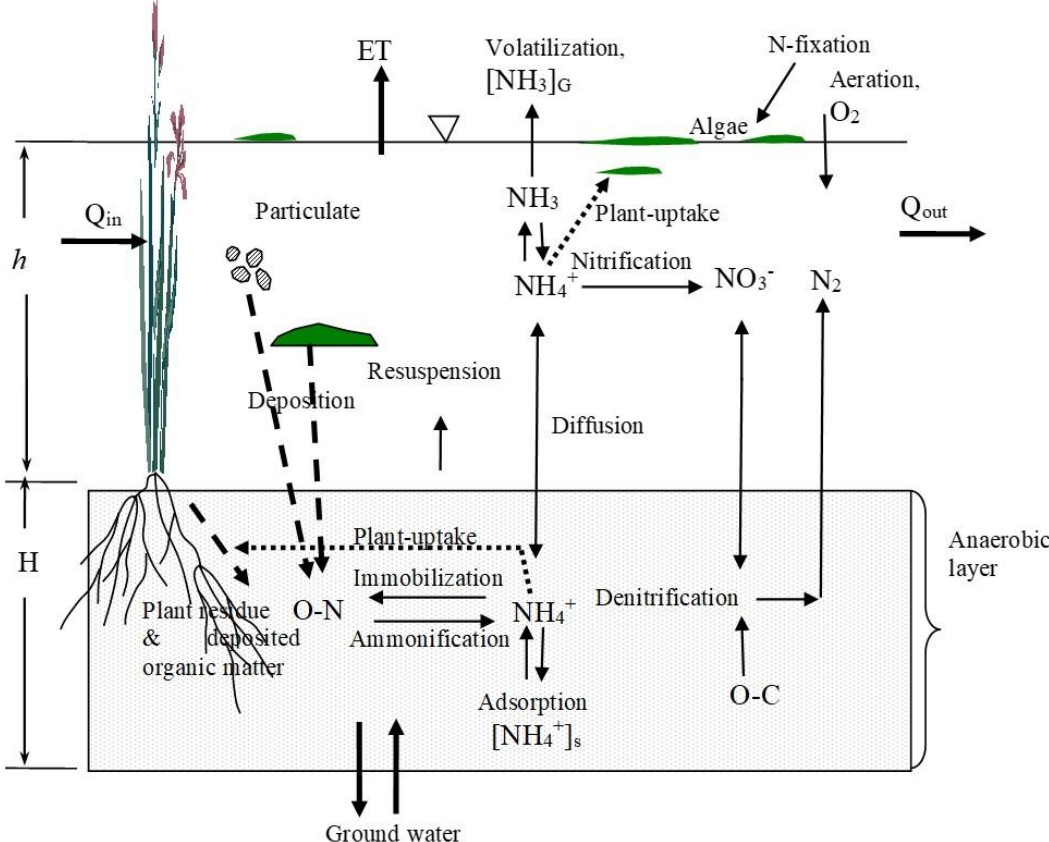

**Figure 2.** Schematic diagram of nitrogen processes in wetlands: water column and soil layer: Figure modified and reproduced from Hantush, et al. [11].

The main processes in this module include mineralization of organic matter to ammonia and phosphate, hydrological transport of nutrients and sediment, nutrient retention, uptake and removal (denitrification, volatilization and burial) in flooded wetlands. The implemented model partitions a wetland into two basic compartments, the water column (free-water) and wetland soil layer. The soil layer is considered to be fully anaerobic. Resuspension of particulate matter and advective and diffusive mass exchange of dissolved constituents are the processes that constitute interactions between the free-water and soil compartments. Agricultural and urban runoff and groundwater discharge constitute the main sources of ammonia and nitrate in the wetland water column. Secondary sources for these constituents are mineralization of suspended organic nitrogen (ON), diffusion, sediment resuspension and atmospheric depositions. Loss of $NH_4$ and production of $NO_3$ occurs in the aerobic water column, whereas $NO_3$ loss by denitrification is limited to the anaerobic soil layer. Ammonia ($NH_3$) volatilization to the atmosphere is a significant loss pathway for N, specifically in soils with a high pH and considered as a N loss pathway in the model.

Primary sources of inorganic phosphorus (P) in wetland soil and water include hydrologically imported P through runoff and decomposition of organic matter. The processes of hydrologic transport, settling, resuspension and diffusion apply to inorganic P in the wetland water. Loss pathways for P

include sedimentation and resuspension of particulate P (organic and adsorbed P) with no gaseous loss pathways. Binding of mineral P (orthophosphate) to organic matter and sediment particles is modeled here using a linear adsorption isotherm.

Similar to the WetQual model, it is assumed that concentrations of various nutrients are uniform in each layer (complete mixing in water and sediment). Most N reactions in the model (mineralization, nitrification, denitrification, etc.) are considered though first-order reaction kinetics. The next section presents the underlying mass balance relationships in the model.

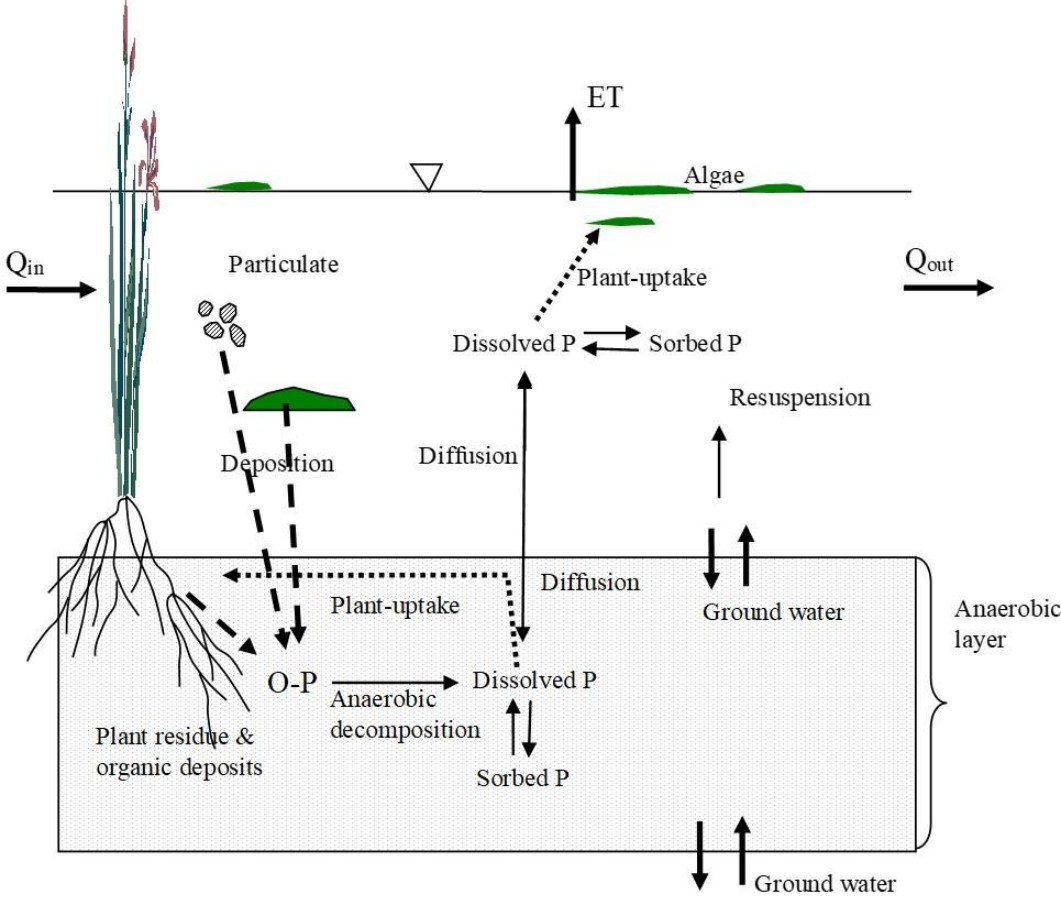

**Figure 3.** Schematic diagram of phosphorus processes in wetlands, including the water column and reduced soil layer. Figure modified and reproduced from Hantush, et al. [11].

### 2.2.1. Nitrogen Dynamics

Water Column:

$$\phi_w \frac{d(V_w N_{ow})}{dt} = Q_i N_{owi} + a_{na} k_{da} a + a_{na} k_{db} f_{bw} b - \phi_w V_w k_{mw} N_{ow} - v_s \phi_w A N_{ow}$$
$$+ v_s \phi_w A (N_{or} + N_{os}) - Q_o N_{ow} + A f_{Sw} S \tag{5}$$

$$\phi_w \frac{d(V_w N_{aw})}{dt} = Q_i N_{awi} + i_p A N_p + \phi_w V_w f_N k_{nw} N_{aw} + \beta_a A (N_{as} - N_{aw}) + F_{N_a g}^w$$
$$- k_v \phi_w A (1 - f_N) N_{aw} + \phi_w V_w k_{mw} N_{ow} - Q_o N_{aw} - f_{aw} a_{na} k_{ga} a + A q_a \tag{6}$$

$$\phi_w \frac{d(V_w N_{nw})}{dt} = Q_i N_{nwi} + i_p A N_{np} + \phi_w V f_N k_{nw} N_{aw} + \beta_a A (N_{ns} - N_{nw}) + F_{N_a g}^w$$
$$- f_{nw} a_{na} k_{ga} a - Q_o N_{nw} + + A q_n \tag{7}$$

where $N_{ow}$ is particulate organic nitrogen (ON) concentration in free water ($ML^{-3}$); $N_{aw}$ = total ammonical-N (NH$_3$+NH$_4$)-N concentration in free water ($ML^{-3}$); $N_{nw}$ is NO$_3$-N concentration in free water ($ML^{-3}$); a is mass of free floating plants (M *Chl a*); b is mass of rooted plants (M *Chl a*); $N_{owi}$, $N_{awi}$

and $N_{nwi}$, respectively, are concentrations of organic N, total ammonical N and NO$_3$-N in incoming inflow (ML$^{-3}$); $N_{as}$ and $N_{ns}$, respectively, are pore-water concentrations of total ammonical N and NO$_3$-N anaerobic soil layer (ML$^{-3}$); $N_{os}$ is concentration of ON in wetland soil (ML$^{-3}$); $N_{ap}$ and $N_{np}$, respectively, are concentrations of total ammonical N and NO$_3$-N in precipitation (ML$^{-3}$); $q_a$ and $q_n$, respectively, are dry depositional rates of total ammonical N and NO$_3$-N (ML$^{-2}$T$^{-1}$); $v_s$ is effective settling velocity (LT$^{-1}$); $v_r$ is resuspension rate (LT$^{-1}$); $S$ is rate of N fixation by microorganisms (ML$^{-2}$T$^{-1}$); and $f_N$ is the fraction of total ammonical N as NH$_4$. All other related physical, biochemical, reaction and physiological parameters are defined in Table 1.

**Table 1.** Wetland model parameter definitions.

| | Definition (unit) |
|---|---|
| $a_{na}$ | gram of nitrogen per gram of chlorophyll-a in plant/algae |
| $a_{oc}$ | gram of oxygen produced per gram of organic carbon synthesized (=2.67) |
| $a_{pa}$ | gram of phosphorus per gram of Chlorophyll-a |
| $a_{pn}$ | phosphorus to nitrogen mass ratio produced by mineralization of POM (=1.389) |
| $f_{a1}$ | fraction of mineral nitrogen plant uptake as ammonia-N in the soil aerobic layer |
| $f_{a2}$ | fraction of mineral nitrogen plant uptake as ammonia-N in the soil anaerobic layer |
| $f_{aw}$ | fraction of mineral nitrogen plant uptake as ammonia-N in free water |
| $f_{bs}$ | fraction of rooted plant biomass below soil-water interface (within soil layer) |
| $f_{bw}$ | fraction of rooted plant biomass above soil-water interface |
| $f_{nw}$ | fraction of mineral nitrogen plant uptake as nitrate-N in free water |
| $f_{n2}$ | fraction of mineral nitrogen plant uptake as nitrate-N in the anaerobic layer |
| $f_N$ | fraction of total ammonia nitrogen ($[NH_4^+] + [NH_3]$) as $NH_4^+$ |
| $f_r$ | fraction of rapidly mineralizing particulate organic matter |
| $f_s$ | fraction of slowly mineralizing particulate organic matter |
| $f_{Sw}$ | fraction of nitrogen fixation in water |
| $k_{da}$ | death rate of free-floating plants (T$^{-1}$) |
| $k_{db}$ | death rate of benthic and rooted plants (T$^{-1}$) |
| $k_{dn}$ | denitrification rate in anaerobic soil layer (T$^{-1}$) |
| $k_{ga}$ | growth rate of free-floating plants (T$^{-1}$) |
| $k_{gb}$ | growth rate of benthic and rooted plants (T$^{-1}$) |
| $k_{mr}$ | first-order rapid mineralization rate in wetland soil (T$^{-1}$) |
| $k_{ms}$ | first-order slow mineralization rate in wetland soil (T$^{-1}$) |
| $k_{mw}$ | first-order mineralization rate in wetland free water (T$^{-1}$) |
| $k_{nw}$ | first-order nitrification rate in wetland free water (T$^{-1}$) |
| $k_s{}^*$ | maximum first-order nitrification rate in wetland soil (T$^{-1}$) |
| $k_v$ | volatilization mass transfer velocity (LT$^{-1}$) |
| $k_{nw}{}^*$ | maximum first-order nitrification rate in wetland free water (T$^{-1}$) |
| $K_d$ | ammonium ion distribution coefficient (L$^3$M$^{-1}$) |
| $K_o R_s$ | oxygen reaeration mass-transfer velocity (LT$^{-1}$)total ammonia retardation factor in wetland soil |
| $r_{c,chl}$ | carbon mass ration in chlorophyll a |
| $r_{on,m}$ | gram of oxygen consumed per gram of organic nitrogen mineralized (=15.29) |
| $r_{on,n}$ | gram of oxygen consumed per gram of total ammonium nitrogen oxidized by nitrification (=4.57) |
| $\alpha, \eta$ | empirical parameters in the relationship relating oxygen liquid-film transfer velocity to wind speed |
| $\beta_a, \beta_n$ | diffusive mass-transfer rate of dissolved N between wetland water and soil layer (LT$^{-1}$) |
| $\beta_p$ | diffusive mass-transfer rate of dissolved phosphorus between wetland water and soil layer (LT$^{-1}$) |
| $\lambda_s, \lambda_w$ | empirical coefficients limiting nitrification in soil ($\lambda_s$) and free water ($\lambda_s$) based on oxygen availability |

In Equation (2), the term (1-$f_N$), which is the fraction of (NH$_3$+NH$_4$)-N as NH$_3$, appears because volatilization is limited to this fraction. Refer to Hantush, et al. [11] for derivation of model variables and values of model constants.

Anaerobic Soil Layer:

$$V_s \frac{dN_{or}}{dt} = f_r a_{na} k_{db} f_{bs} b + f_r v_s \phi_w A N_{ow} - v_r \phi_w A N_{os} - V_s k_{mr} N_{os} - v_b A N_{os} + f_r (1 - f_{Sw} S) A \quad (8)$$

$$\phi V_s R_s \frac{dN_{as}}{dt} = -A\beta_a (N_{as} - N_{aw}) - \phi A v_b (N_{as} - N_{aw}) + V_2 k_{mr} N_{ow} + V_2 k_{ms} N_{ow} - f_{a2} a_{na} k_{gb} f_2 b \quad (9)$$

$$\phi V_s \frac{dN_{as}}{dt} = -A\beta_n (N_{nw} - N_{ns}) - \phi V_2 k_{dn} N_{ns} - \phi A v_b (N_{ns} - N_{nw}) - f_{n2} a_{nd} k_{gb} f_2 b \quad (10)$$

where $N_{os}$ is defined above; $v_b$ is burial velocity ($LT^{-1}$); $Vs = H\,A$ is volume of active sediment layer ($L^3$); and $H$ is thickness of active soil layer (L) (refer to Table 1 for definition of all other physical and biochemical parameters/coefficients).

### 2.2.2. Sediment Dynamics

Sediment transport and fate in wetland water may be described by this equation

$$\phi_w \frac{d(V_w m_w)}{dt} = Q_i m_{wi} - v_s \phi_w A m_w + v_r \phi_w A m_s - Q_o m_w \tag{11}$$

Mass balance in the active soil compartment is given by

$$V_s \frac{dm_s}{dt} = v_s \phi_w A m_w - v_r \phi_w A m_s - v_b A m_s \tag{12}$$

where $m_w$ is sediment concentration in free water ($ML^{-3}$); $m_{wi}$ is sediment concentration in incoming flow ($ML^{-3}$); $m_s = (1 - \phi)\rho_s$ is wetland soil bulk density ($ML^{-3}$); and $\rho_s$ is soil particle density ($ML^{-3}$).

### 2.2.3. Phosphorus Dynamics

Water Column:

$$\frac{d(V_w P_w)}{dt} = Q_i P_{wi} - v_s F_{sw} m_w \phi_w A P_w + v_r \phi_w A f_{ss} m_s P_s - a_{pa} k_{ga} a + V_w a_{pn} k_{mw} N_{ow}$$
$$+ \beta_p A (F_{da} P_s - F_{dw} P_w) - Q_o P_w \tag{13}$$

Anaerobic Soil Layer:

$$V_s \frac{dP_s}{dt} = f_2 \phi_w A v_s m_w F_{sw} P_w - \phi_w A v_r m_s f_{ss} P_s + V_s a_{pn} k_{mr} N_{ow} + V_s a_{pn} k_{ms} N_{os}$$
$$+ \beta_p A (F_{da} P_w - F_{ds} P_w) - a_{pa} k_{gb} f_2 b \tag{14}$$

where $P_w$ is total inorganic P concentration in free water ($ML^{-3}$); $P_{wi}$ is inflow total inorganic P concentration ($ML^{-3}$); Ps is total P concentration in anaerobic layer ($ML^{-3}$); $F_{d,w}$ is dissolved fraction of total inorganic P in free water; $m_w\,F_{s,w}$ is a sorbed fraction of total inorganic P in free water $(1 - F_{d,w})$; $m_s\,F_{s,s}$ is a sorbed fraction of total inorganic P in aerobic layer; and $K_{s,2}$ is distribution coefficient in reduced wetland soil ($L^3 M^{-1}$).

### 2.3. Study Area

An enhanced APEX model was tested in a study wetland with approximately two years of monitored outflow and water quality data, described thoroughly by Jordan, et al. [16]. The study site is a small restored wetland located on Kent Island, Maryland (Figure 4). Several studies have been conducted to understand wetland hydrology for this region using a modeling approach [17–20]. During the two-year sampling period, the study wetland had an average area of 1.3 ha and drained to a 14-ha watershed that was mainly covered by crop fields (82%) and forest (18%). The study wetland was restored from an artificially drained cropland by the Chesapeake Wildlife Heritage with the intention to provide wildlife habitat and improve the quality of runoff from surrounding crop fields. Emergent vegetation covered roughly 90% of the wetland surface during the growing season and 10% during the non-growing season. Water entered the wetland through ditches draining surface runoff from the surrounding catchment and outflowed via a standpipe connected to a 120° V-notch weir. The entire 1.3-ha wetland area was submerged and lacked well-defined flow channels when water was deep enough to flow out of the weir. An impermeable layer of clay was added within 0.5 m of the soil surface during wetland restoration. This layer blocked groundwater exchanges and infiltration.

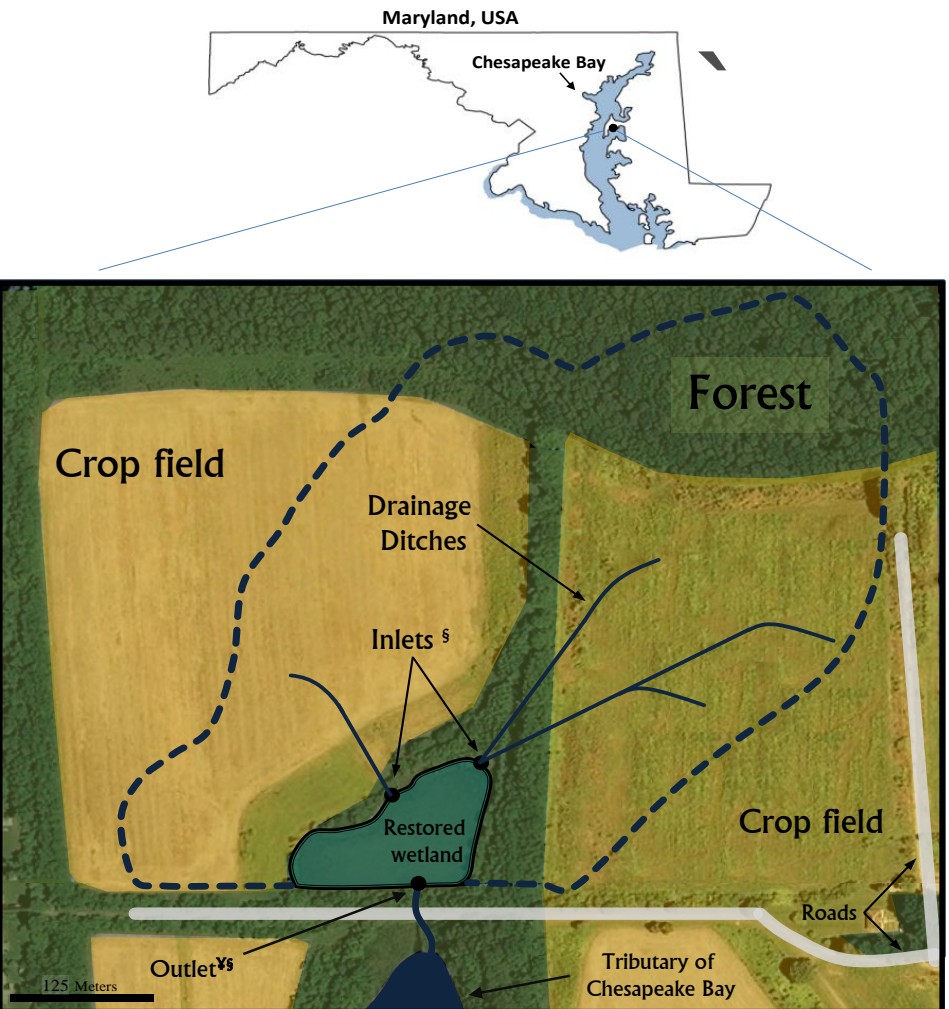

¥ Outflow structure consisted of a standpipe connected to a 120° V-notch weir.
§ Automated samplers were installed to sample water entering and leaving the wetland.

**Figure 4.** Study wetland located on Kent Island, MD. The watershed draining to the wetland is outlined by dashed lines (adopted from Sharifi, et al. [15]).

Wetland bathymetry and boundary data were acquired through field surveys and mapping that was done by researchers with the Smithsonian Environmental Research Center at the time of the study [16]. Automated instruments were used to measure unregulated water inflows and to sample water entering and leaving the wetland from 8 May 1995 through 12 May 1997. Weekly (typically 5 to 8 days) flow averaged $NO_3$, $(NH_3 + NH_4)$-N, ON, inorganic P and total suspended sediment (TSS) concentrations in runoff were available from Jordan, et al. [16]. Details of data collection and analysis can be found in Jordan, et al. [16].

In short, Jordan, et al. [16] produced volume-weighted composite samples by installing data logger controlled pumps that collected separate water samples of inflowing and outflowing water, in volumes proportional to respective flow rates. The composite samples were collected weekly and were brought into the laboratory for analysis. To convert weekly average concentrations reported by Jordan, et al. [16] into daily values, we assumed that concentrations were constant over the given weekly periods. The dataset also contained periods where data were missing. We reconstructed the records during such periods by taking averages of the last available measurement before the gap and the first available measurement after the gap.

### 2.4. Hydrologic Calibration

Although the APEX model is not appropriately equipped to model wetland hydrology, we found that the reservoir component within APEX can be fine-tuned to simulate hydrology of ponded wetlands. The reservoir module uses a simple mass balance equation to calculate daily volume of stored water. Water flow is controlled via principal and emergency spillways. Volume and surface area of stored water are related via an exponential relationship. Surface area in turn affects daily evapotranspiration and infiltration. Details of the module can be found in APEX theoretical documentation [13]. Fine tuning was achieved by manipulating a handful of model parameters involving reservoir elevation, volume and area at spillway level and time to release flood storage. Calibration was performed manually through direct comparison of simulated volume, inundated area and outflow against field observations.

### 2.5. Model Uncertainty and Performance Assessment

Model assessment applied in this study implements a Generalized Likelihood Uncertainty Estimation (GLUE) method [21], where 50,000 statistically independent parameter sets are generated randomly from prior distributions extracted from literature. Monte Carlo (MC) simulations were performed to yield an ensemble of 50,000-time series for modeled constituent loadings of sediments, ON, $NO_3$, $(NH_3 + NH_4)$-N and $PO_4$. Nash–Sutcliffe efficiency (NSE) was used as a performance measure to evaluate the goodness-of-fit between model-predicted loadings and observed data for each MC simulation. Simulated results were sorted based on their performance and the top 1% of runs were selected to represent the behavioral parameter set and depict model uncertainty. The enhanced APEX model was run at a daily time step from 8 May 1995 through 12 May 1997 and assessed for five water quality variables (sediments, ON, $NO_3$, $(NH_3 + NH_4)$-N and $PO_4$). Daily simulations were aggregated into weekly values to compare with weekly observation values. Mass balance error (MBE $= (\sum Observed\ load - \sum Simulated\ load)/(\sum Observed\ load)$ and $R^2$ were used in addition to NSE for assessing model performance.

As mentioned earlier, APEX was first calibrated manually for hydrologic variables (i.e., outflow, volume and inundated area). GLUE uncertainty assessment was performed exclusively for the newly added module (hydrologic model variables were kept untouched between MC simulations).

## 3. Results

### 3.1. Wetland Hydrology

Monthly APEX simulations of wetland hydrologic variables (flow, volume and inundated area) are compared with field observations in Figure 5. As depicted in the Figure, APEX can be fine-tuned to simulate wetland hydrology fairly well. APEX simulations of monthly water volume exhibited a fairly good overall performance (NSE = 0.68 and $R^2$ = 0.89). Model performance decreased during the first dry/wet signal between May-1995 and Nov-1995. During this period observed water volume seems to drop and replenish at steeper rate compared with simulated volume. The same pattern is observed for wetland inundated area during the first dry/wet period. APEX tends to slightly underestimate volume for the rest of the period (Dec-1995–May-1997). APEX shows much less sensitivity in capturing area variations and has a weaker fit compared with volume and outflow predictions (NSE = 0.15, $R^2$ = 0.9). Despite all facts stated above, APEX captured wetland outflow quite well (NSE = 0.93, $R^2$ = 0.93). Outflow is perhaps the most important factor amongst the hydrologic variables for biogeochemical modeling, as outflow is strongly tied to the transport of sediment, N and P.

Natural and constructed wetlands do not often contain clearly defined outflow channels or structures, which make physical hydrologic modeling for such wetlands a difficult task. This application shows that the APEX reservoir module has the capacity to simulate ponded wetland hydrology. Although this approach is not without pitfalls—as some level of calibration is

required—parameters controlling reservoir hydrology in APEX can be either extracted from bathymetry data or transferred from previous studies.

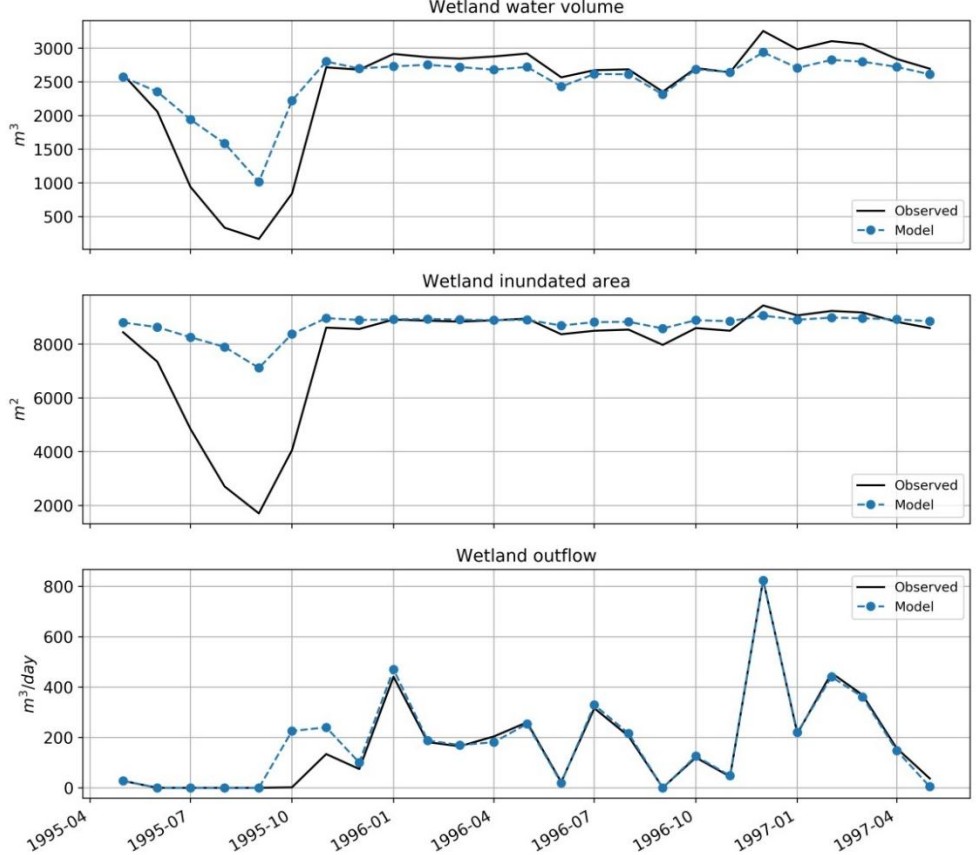

**Figure 5.** Comparison of monthly simulated and observed wetland volume (NSE = 0.68, $R^2$ = 0.89), inundated area (NSE = 0.15, $R^2$ = 0.9) and outflow (NSE = 0.93, $R^2$ = 0.93).

### 3.2. Wetland Biogeochemistry

Model performance measures (NSE, $R^2$ and MBE) for the behavioral simulations (top 1% or 50,000 runs, n = 5000) are presented in Figure 6. Figure 7 presents model uncertainty bands for behavioral model runs, in comparison with observed weekly loadings and best model run (holding the highest NSE) for predicted water quality constituents. Both figures in conjunction give us a clear understanding of model behavior and model performance over the simulation period.

#### 3.2.1. Sediment

The model showed good behavioral performance on sediment predictions (0.53 < NSE < 0.70, 0.71 < $R^2$ < 0.74, −46.14 < MBE < −21.4). The best performance has NSE = 0.70 which is generally considered favorable. The model consistently underestimates sediment load with negative MBE for all behavioral simulations. Model uncertainty on sediment predictions (green band on top panel of Figure 7) appears to be narrow for the most part, except for periods of peak loading. This pattern (high uncertainty during peak flow) can also be observed on model predictions of most other constituents. The model had trouble capturing peak sediment outflows at three points in time (indicated by red arrows on sediment panel of Figure 7). This indicates that the actual wetland is flashier than its modeled representation. This can be addressed by adjusting parameters associated with movement of sediment, that is, settling and resuspension rates in the model.

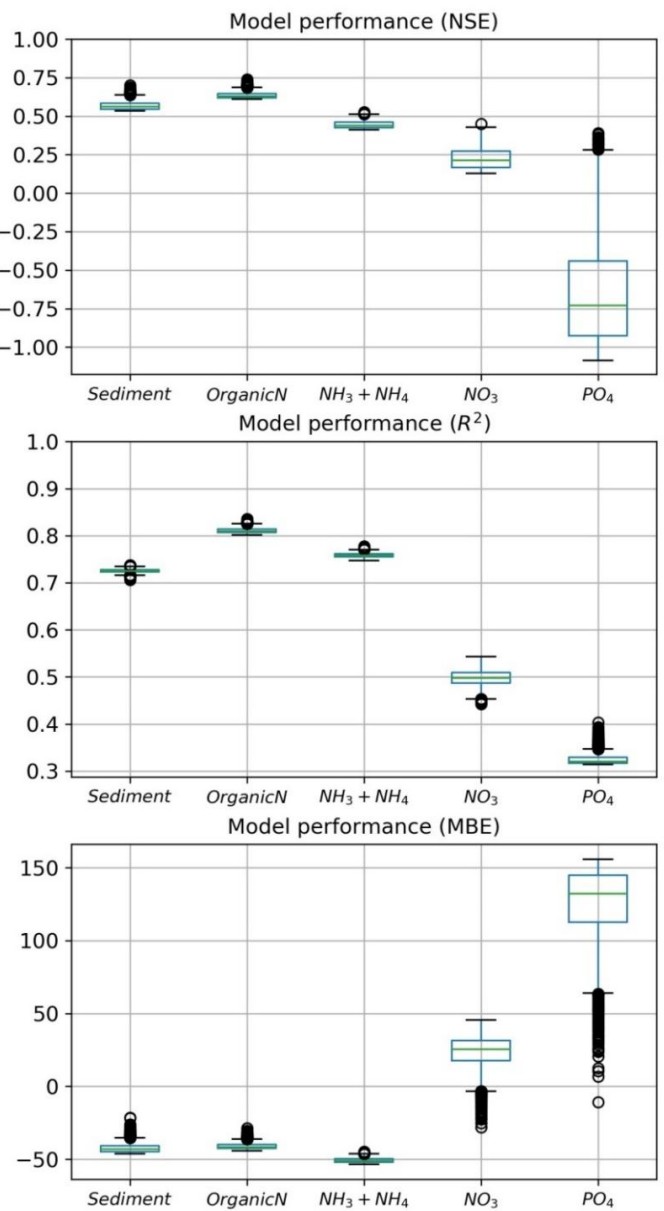

**Figure 6.** Model performance for behavioral runs: NSE, R$^2$ and MBE.

### 3.2.2. Nitrogen

The model performed well in predicting ON loadings (0.61 < NSE < 0.74, 0.8 < R$^2$ < 0.84, −44.3 < MBE < −28.63) where best performance exhibited an NSE = 0.74. This is the highest performance the model reported among all constituents. Performances for (NH$_3$ + NH$_4$)-N are fairly acceptable as well (0.41 < NSE < 0.53, 0.75 < R$^2$ < 0.78, −53.35 < MBE < −44.73). ON and (NH$_3$ + NH$_4$)-N model predictions both have fairly narrow uncertainty bands and both underestimate loadings with negative MBEs for all behavioral simulations. Once more, field observations describe a flashier wetland compared with the model in terms of ON and (NH$_3$ + NH$_4$)-N loadings. One possible explanation is that more ON is being settled and buried in the model (similar to sediment), therefore becoming less available for ammonification (hence underestimation of NH$_3$ + NH$_4$)-N and wetland export. As observed with sediment, some peak ON loadings are missed by the model (indicated with red arrow on Figure 7). Variation in observed (NH$_3$ + NH$_4$)-N loading is higher than all other constituents and the model has a hard time matching this variability.

$NO_3$ predictions are the weakest of all N constituents ($0.13 < NSE < 0.45$, $0.44 < R^2 < 0.54$, $-28.03 < MBE < 45.62$); however, all behavioral simulations yielded positive NSE and the best model has $NSE = 0.45$. Unlike previous instances (sediment, ON, $(NH_3 + NH_4)$-N), most $NO_3$ behavioral models overestimate loadings and MBE for $NO_3$ model predictions has a wide spread (see Figure 6, MBE panel). Model uncertainty (spread) is wider compared with ON and $(NH_3 + NH_4)$-N and seems widest at times of peak loading. It is worth mentioning that $NO_3$ concentrations in the model are directly tied with hydrology (through hydrologic export and groundwater exchange), $(NH_3 + NH_4)$-N concentration (through nitrification) and indirectly tied to ON concentration (through ammonification). Any uncertainty associated with hydrology, ON and $(NH_3 + NH_4)$-N predictions will in turn effect $NO_3$ predictions. Therefore, it is only natural to expect lower performance and higher uncertainty for $NO_3$.

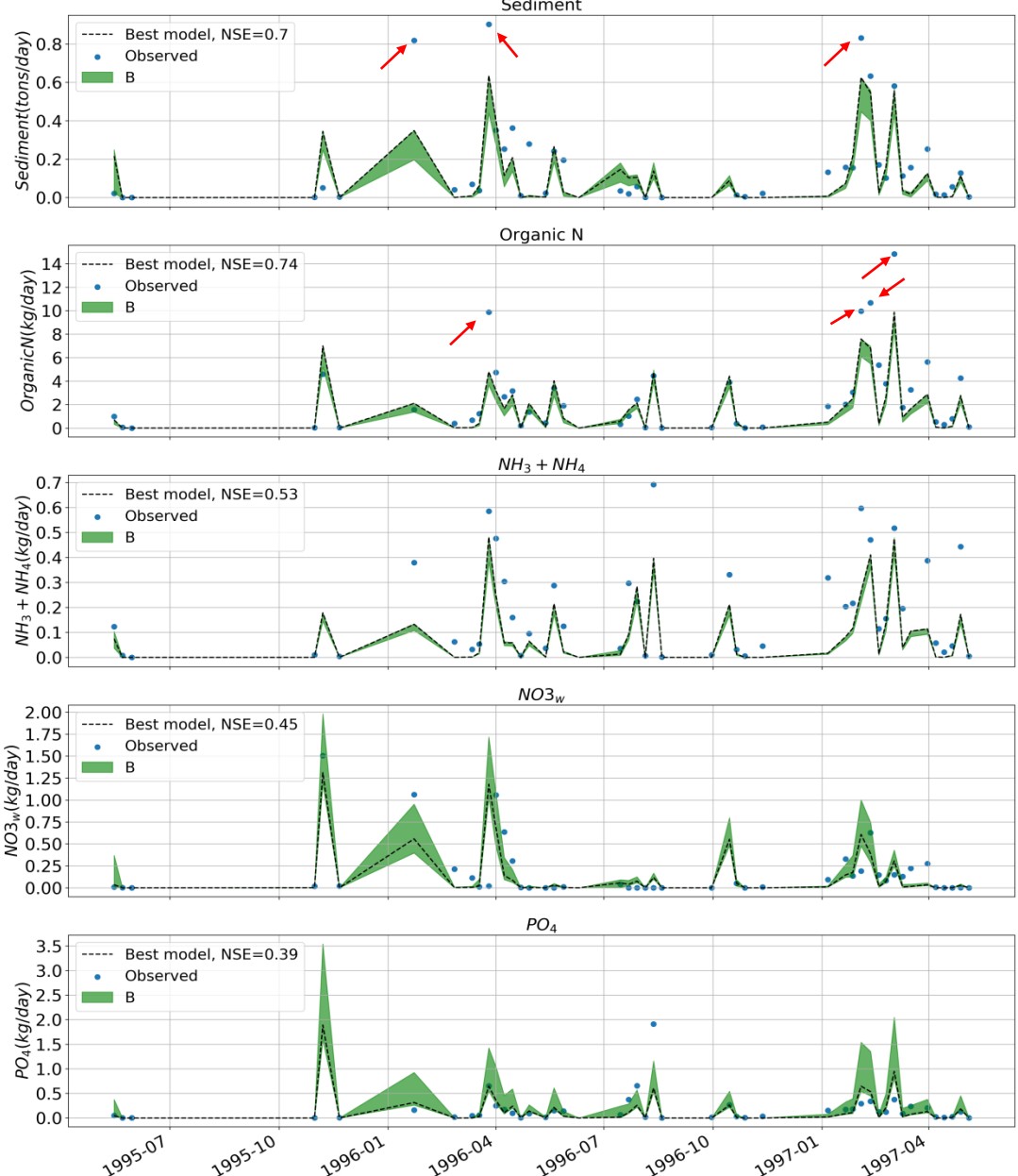

**Figure 7.** Model uncertainty bands for behavioral simulations in comparison with best simulation (highest NSE) and weekly observed loadings for sediment, ON, $(NH_3 + NH_4)$-N, $NO_3$ and $PO_4$.

### 3.2.3. Phosphorus

Model performance for $PO_4$ was the weakest amongst all water quality constituents reported in this paper ($-1.09 < NSE < 0.39$, $0.31 < R2 < 0.4$, $-10.82 < MBE < 155.8$). Although best performance has $NSE = 0.39$, the majority of behavioral performances resulted in negative NSE (median $NSE = -0.72$). Behavioral model predictions for $PO_4$ have a wide MBE spread and almost all models overestimate $PO_4$ export. Unlike earlier examples, in this instance, the model acts flashier than what observed data show in terms of $PO_4$ export. Capturing $PO_4$ variability is a challenging task, as $PO_4$ concentrations in the model are tied to sediment and ON concentrations, in addition to hydrology (see Section 2.2.3). Any uncertainty associated with the named variables will introduce uncertainty in $PO_4$ predictions. It is generally understood that capturing $PO_4$ variability in wetlands/aquatic environments is more challenging than N related constituents.

### 4. Discussion and Limitation

Although our wetland module improved the ability of the APEX model to simulate biogeochemical processes within a wetland, the enhanced APEX possesses limitations that stem from underlying assumptions, including (1) assumption of complete mixing of nutrients and sediment within the wetland waterbody and sediment layer, (2) omission of the thin aerobic layer that forms on top of the wetland anaerobic sediment layer and (3) no consideration of the thin aerobic coating of soil around plant roots with aerenchymatic stems. These limitations can be addressed in future generations of the model. However, it is noteworthy to mention that for this type of study (i.e., nutrient cycling in wetlands within agricultural environments), forces outside of wetland biogeochemical cycling, such as hydrologic nutrient imports and exports to/from wetlands, exert far greater uncertainties on model predictions than these limitations. In other words, to reduce total model uncertainty for this study, improving the watershed component of the APEX model—which dictates nutrient input to the receiving wetland—and enhancing the hydrological component of the reservoir model—which was interchangeably used for the wetland module—would be more effective than focusing more on biogeochemical cycling within the wetland.

Improved availability of continuous *in-situ* wetland nutrient datasets will also reduce uncertainty of wetland biogeochemical models. The field dataset presented in this study, first published by Jordan et al. [16], is one of the few existing datasets of its type. Collection of field hydrologic and water quality data is expensive, time consuming and technically challenging. Increased availability of *in-situ* real-time water quality monitoring devices will facilitate the collection of these datasets.

Sampling frequency likely contributed to increased model uncertainty. Sampling methodology used by Jordan, et al. [16] included collecting volume-weighted composite samples weekly. Low-frequency field sampling has been found to increase uncertainty of *in-situ* datasets by excluding key hydrologic processes that operate at high temporal intervals [22,23]. Therefore, sampling data might not reflect day to day variability in nutrient imports and exports, increasing prediction uncertainty.

### 5. Summary and Conclusions

In this study, a modified version of the WetQual model was added to an existing reservoir sub-routine within APEX. This new module enabled APEX to simulate mineralization, nutrient retention, hydrologic transport and accurate removal processes (e.g., denitrification) in wetlands. The performance of an enhanced model was evaluated at a wetland on the Eastern Shore of the Maryland using five observed water quality variables (e.g., sediment, ON, $NO_3$, $(NH_3 + NH_4)$-N and $PO_4$). The APEX model was first calibrated for hydrologic variables (wetland inundated volume and area, wetland outflow). Model assessment was performed by implementing GLUE methodology. 50,000 model runs were performed using randomly generated parameter sets and the top performing (1%) models were selected as behavioral models.

The amended model showed good behavioral performance on sediment predictions ($0.53 < \text{NSE} < 0.70$, $0.71 < R^2 < 0.74$, $-46.14 < \text{MBE} < -21.4$) with narrow behavioral uncertainty except for periods of peak loading. Organic N loadings were substantially captured by the model ($0.61 < \text{NSE} < 0.74$, $0.8 < R^2 < 0.84$, $-44.3 < \text{MBE} < -28.63$). Model performance for $(NH_3+NH_4)$-N were in the acceptable range ($0.41 < \text{NSE} < 0.53$, $0.75 < R^2 < 0.78$, $-53.35 < \text{MBE} < -44.73$). Organic N and $(NH_3+NH_4)$-N model predictions both showed narrow uncertainty bands and both underestimated loadings. $NO_3$ model predictions were the weakest between N constituents ($0.13 < \text{NSE} < 0.45$, $0.44 < R^2 < 0.54$, $-28.03 < \text{MBE} < 45.62$). This was not unexpected, as model uncertainties linked to hydrologic and ON, $(NH_3 + NH_4)$-N predictions are directly passed down to $NO_3$ predictions through association.

Model performance for $PO_4$ was the weakest amongst all water quality constituents ($-1.09 < \text{NSE} < 0.39$, $0.31 < R^2 < 0.4$, $-10.82 < \text{MBE} < 155.8$) and majority of behavioral performances had negative NSE. Uncertainties associated with $PO_4$ predictions were expected to be higher than other constituents, as $PO_4$ concentrations in the model are tied to sediment and ON concentrations, in addition to hydrology.

Although additional efforts to improve the model algorithm and validate model performance in a range of wetlands are imperative, this study demonstrates that APEX can be modified to better reflect physical processes pertaining to circulation of sediment, nitrogen and phosphorus in inundated wetlands. The new module developed as part of this study enables the APEX model to evaluate the effects and effectiveness of wetland restoration and conservation by better quantifying improvements to water quality provided by wetlands.

**Author Contributions:** A.S. developed methodology and coding for the model; A.S. and S.L. performed most analysis and writing; G.W.M. and M.W.L. helped with writing and supervised the project; J.J., A.M.S. and M.C.R. were instrumental in writing and data analysis.

**Funding:** This research was funded by the US Department of Agriculture (USDA) Natural Resources Conservation Service (NRCS) Conservation Effects Assessment Project (CEAP) grant number (67-3A75-17-473).

**Acknowledgments:** We sincerely appreciate the valuable time and effort of the reviewers, which resulted in significant improvement to this publication. We also wish to thank the Journal Editor Board for their help and patience throughout the review process.

**Conflicts of Interest:** The authors declare no conflict of interest. The funding sponsors had no role in the design of the study, in the collection, analyses or interpretation of data, in the writing of the manuscript or in the decision to publish the results.**Disclaimer**: The findings and conclusions in this article are those of the author(s) and do not necessarily represent the views of the U.S. Fish and Wildlife Service.'

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
