# Peer review of "Enhancement of Agricultural Policy/Environment eXtender (APEX) Model to Assess Effectiveness of WetlandWater Quality Functions"

_water, doi:10.3390/w11030606_

Reviewer 1 Report

The authors presented a modified version of the APEX model for wetlands. The model was tested based on a realistic case study and the authors evaluated the water balance, sediment, organic N, NO3, NH4, PO4 simulations. I have to note that the simultaneous simulation of all these peramaters and the comparison of the results versus real measurements is one of the most challenging cases in ecological modelling. Considering that, the model presented an adequate overall model performance. My main concern is that the authors considered the bottom layer anaerobic and excluded completely the existence of a thin aerobic layer. They also claim that it was neglected on purpose due to negligible differences in the results (lines 154-155). I believe that this significantly reduces the general applicability of the model. Moreover, in the majority of wetlands with aquatic plants which have aerenchymatic stems, oxygen is transfered to the root layer creating aerobic conditions around the roots. I don't know the exact conditions of the tested wetland, but this model version is probably more representative for the specific wetland rather for general use.  My recommendation is to split the results and discussion section in order to provide a more rebust description of the limitations of the model and further recommendations for improvements.

Author Response

Dear a reviewer,

Thanks for your valuable comments on the manuscript. I have revised the manuscript according to comments. My responses are summarized in the attached document.

Best,

Sangchul

Reviewer 2 Report

This is an interesting evaluation of a potential improvement to the APEX model to include wetlands and potentially account for the nutrient cycles. A few minor suggestions are provided;

The APEX and WetQual models are not well-known internationally. Perhaps the authors could indicate similar models (if/where they exist) that international readers might be familiar with and could compare to?

It is acknowledged that the primary data has been drawn from the referenced Jordan paper, but for the benefit of the reader, it would be valuable to expand a little more on the sample collection and analysis process.

The amended model appears to have merit with many of the predicted outcomes in close alignment with the observed data. It is noted, however, that where the predicted and observed differ (Figure 5 & 7), the amount can be significant. The paper suggests that this might be because the wetland is more "flashy" than the model in its hydrological response. Acknowledging that I haven't read the Jordan paper, could it be that the sampling methodology has influenced the observations in a way not anticipated by the model? For example, were samples collected shortly after a storm event, where other weekly samples were not influenced by event runoff? Or could it be as a result of the conversion of weekly into daily values, or missing data gaps? Could it be explained by seasonal factors (rainfall, temperature, wind direction contributing airborne NO3? Or could it be simply that environmental variability with these pollutants requires large error bands?

What would the authors recommend for further research to assist in improving the accuracy of the model?

An interesting paper, well-written and detailed.

Author Response

Dear a reviewer,

Thanks for your valuable comments on the manuscript. I have revised the manuscript according to comments. My responses are summarized in the attached document.

Best,

Sangchul

Round  2

Reviewer 1 Report

no further comments